# Vertebrate Cell Differentiation, Evolution, and Diseases: The Vertebrate-Specific Developmental Potential Guardians *VENTX*/*NANOG* and *POU5*/*OCT4* Enter the Stage

**DOI:** 10.3390/cells11152299

**Published:** 2022-07-26

**Authors:** Bertrand Ducos, David Bensimon, Pierluigi Scerbo

**Affiliations:** 1LPENS, PSL, CNRS, 24 rue Lhomond, 75005 Paris, France; 2IBENS, PSL, CNRS, 46 rue d’Ulm, 75005 Paris, France; 3High Throughput qPCR Core Facility, ENS, PSL, 46 rue d’Ulm, 75005 Paris, France; 4Department of Chemistry and Biochemistry, UCLA, Los Angeles, CA 90094, USA

**Keywords:** developmental potential, heterogeneity, competence, pluripotent stem cells, neural crest cells, cancer, vertebrate evolution, *VENTX*/*NANOG*, *POU5*/*OCT4*

## Abstract

During vertebrate development, embryonic cells pass through a *continuum* of transitory pluripotent states that precede multi-lineage commitment and morphogenesis. Such states are referred to as “refractory/naïve” and “competent/formative” pluripotency. The molecular mechanisms maintaining refractory pluripotency or driving the transition to competent pluripotency, as well as the cues regulating multi-lineage commitment, are evolutionarily conserved. Vertebrate-specific “Developmental Potential Guardians” (vsDPGs; i.e., *VENTX*/*NANOG, POU5*/*OCT4*), together with MEK1 (MAP2K1), coordinate the pluripotency *continuum*, competence for multi-lineage commitment and morphogenesis in vivo. During neurulation, vsDPGs empower ectodermal cells of the neuro-epithelial border (NEB) with multipotency and ectomesenchyme potential through an “endogenous reprogramming” process, giving rise to the neural crest cells (NCCs). Furthermore, vsDPGs are expressed in undifferentiated-bipotent neuro-mesodermal progenitor cells (NMPs), which participate in posterior axis elongation and growth. Finally, vsDPGs are involved in carcinogenesis, whereby they confer selective advantage to cancer stem cells (CSCs) and therapeutic resistance. Intriguingly, the heterogenous distribution of vsDPGs in these cell types impact on cellular potential and features. Here, we summarize the findings about the role of vsDPGs during vertebrate development and their selective advantage in evolution. Our aim to present a holistic view regarding vsDPGs as facilitators of both cell plasticity/adaptability and morphological innovation/variation. Moreover, vsDPGs may also be at the heart of carcinogenesis by allowing malignant cells to escape from physiological constraints and surveillance mechanisms.

## 1. Introduction

After fertilization, the vertebrate zygote massively proliferates and gives rise to the blastula/epiblast, an undifferentiated mass of embryonic cells imbued with the potential to acquire all cell fates of the organism (i.e., pluripotency) [1,2]. Once isolated in vitro and upon adequate culture conditions, vertebrate blastula/epiblast cells can be indefinitely maintained and propagate in such undifferentiated state as embryonic stem cells (ESCs) without losing their developmental potential [3,4,5]. Advances in culture conditions actually allow accurate ESCs differentiation and morphogenesis in vitro (i.e., gastruloids, organoids) [6,7]. However, pluripotency in vivo is an ephemeral and transitory phase that is globally dismantled when the three germ layers (i.e., ectoderm, mesoderm, endoderm) are determined and morphogenetic processes (e.g., gastrulation) rearrange embryonic cells along the anterior/posterior (A/P) and dorsal/ventral (D/V) axes of the embryo [8,9,10].

We will integrate knowledge from vertebrate models in order to decipher the conserved molecular logic governing the imperative embryonic cell *continuum* from pluripotency to committed state in vivo. Concomitantly, we sought to propose how and why such molecular logic redeployed during later phases of development could confer cell “evolvability” and “adaptability” in ontogenesis, evolution, and in human diseases such as cancer [11,12]. Vertebrate pluripotent stem cells (PSCs) mature in vivo through a *continuum* of pluripotency state transitions from early blastula/epiblast refractory state (i.e., resistant to “differentiating” cues, “refractory pluripotency”) to late blastula/epiblast competent state (i.e., competent to respond to “differentiating” cues, “competent pluripotency”) (Figure 1A) [9,10,13,14].

The molecular mechanisms governing such a *continuum* are relevant to understanding canalization in vivo, and to translational approaches targeting the obtention of a given cell type. Human orthologs of VENTX and NANOG (*VENTX*/*NANOG* family) and POU5 (*POU5*/*OCT4* family) transcription factors (TF) control such pluripotency *continuum* in vertebrate embryos [14,16,17,18,19,20,21,22,23,24,25,26,27,28]. The conserved ability of these factors to participate in nuclear reprogramming of somatic cells into induced PSCs (iPSCs) further confirm the relevance of these TF in conferring unrestricted cell’s developmental potential [18,19,20,29,30,31,32]. Vertebrate models were crucial in shedding light on the activity of these TF in blastula/epiblast PSCs and during cell commitment to the three germ layers [14,16,17,26,27,28,33,34,35,36]. Expression and function of *VENTX*/*NANOG* and *POU5*/*OCT4* functional homologs is often associated with the maintenance of pluripotency of blastula/epiblast cells, as described in zebrafish, *Xenopus*, and chickens [4,14,16,17,21,22,37,38,39] and in germ line (GL) development [4,40,41,42,43,44], though they are reactivated later in a restricted group of undifferentiated embryonic somatic cells (e.g., neural crest cells, NCCs; posterior neural-mesodermal progenitors, NMPs) [15,45,46,47,48] (Figure 1C).

Since *VENTX*/*NANOG* and *POU5*/*OCT4* are vertebrate-specific innovations that arose in the last common ancestor of extant vertebrates (Figure 1B) [15], we propose to call them vertebrate-specific “Developmental Potential Guardians” (vsDPGs hereafter). Interestingly, whilst cell transplantation of vertebrate donor blastula/epiblast cells (expressing vsDPGs) revealed their capacity to give rise to all somatic cell types of the host organism [2,6,49], comparable studies in invertebrate chordates (i.e., hemichordates, cephalochordates, and urochordates) showed that blastula cells keep a memory of their previous spatial localization while interpreting external stimuli and thus lack *stricto sensu* pluripotency features [6,50,51,52]. This suggest that the molecular origin of vsDPGs may be intimately linked to the rise of a “new and broader competence potential” in cells during vertebrate development.

## 2. Role of Developmental Potential Guardians in Stem Cells Pluripotency

The genes *VENTX*/*NANOG* and *POU5*/*OCT4* of vsDPGs arose in a vertebrate ancestor following whole genome duplication (WDG) and further duplications (and loss) were observed in gnathostomes [15,22,23]. ScRNA sequencing (scRNA-seq) of zebrafish, *Xenopus*, primates (i.e., human and marmoset) embryos highlights the importance of vsDPGs as part of the refractory/naïve pluripotency gene regulatory network (GRN) [24,25,37,38,39,53]. It also confirms functional analyses in vivo and in vitro showing that vsDPGs have crucial functions in maintaining pluripotency and counteracting spontaneous and precocious PSCs differentiation [14,16,17,21,22,26,27,28,33,34,35,36,54,55].

Biochemical and molecular studies in vivo demonstrated that VENTX-POU5 and NANOG-POU5 proteins physically interact as a heterodimer to control transcription of genes required for germ layer commitment, patterning and morphogenesis (Figure 2A) [16,26,27,28,33,34,35,36,54]. VsDPGs heterodimer physically interact also with SMADs proteins (effectors of the TGFβ NODAL/ACTIVIN and BMPs signaling pathways), as well as with CTNNB1 (β-CATENIN)-TCF/TLE proteins (effectors of the WNT signaling pathway) [21,26,27,33,34,35,36,54]. In conjunction with genomic analyses (genome-wide chromatin immunoprecipitation ChIP; gene expression microarrays, RNA-sequencing), these results shed light on the role of vsDPGs as modulators of PSCs competence and actors in the activation of the transcriptional program(s) involved in lineage commitment (e.g., *SOX3*, *GSC*, *HHEX*, *EOMES*, *SOX17*) [21,22,26,27,28,33,34,35,36,49,54,55,56,57,58].

The imperative transition from refractory to competent pluripotency in development lies at the core of complex molecular interaction between vsDPGs and FGF/MAPK signaling pathway. Several studies propose that MEK1 (MAP2K1 kinase, MAPK pathway) is the “universal competence factor” in vertebrate PSCs transitioning from a refractory to a competent state. MEK1 mediates VENTX/NANOG protein degradation by the proteasome through a PEST destruction motif (an amino acid sequence enriched with Proline-P, Glutamate-E, Serine-S and Threonine-T in the VENTX/NANOG N-terminus) thus allowing PSCs to responds to differentiating cues [9,14,59,60,61,62].

Studies in *Xenopus* showed that VENTX protein degradation occurs during PSCs mitosis (anaphase) in a non-polarized asymmetric manner (Figure 2B) [14]. Once PSC enter into mitosis, VENTX localizes on the chromosomes until metaphase, when it detaches from DNA in a MEK1-independent manner. During anaphase, chromosomes of one daughter forming PSC inherit VENTX protein, which re-localizes on DNA, whereas the other daughter forming PSC does not inherit VENTX. Such asymmetric inheritance/distribution of VENTX at anaphase is under the control of MEK1, which regulates unpolarized asymmetric degradation/clearance of VENTX through PEST destruction motif of VENTX at anaphase [14]. The SCF-β-TRCP ubiquitin-mediated global clearance of VENTX allows for germ layer determination at the onset of morphogenesis (i.e., gastrulation) [62]. Thus, asymmetric cell division (ACD) of PSCs in vivo results in a heterogeneous population of PSCs with high-VENTX (refractory) and low-VENTX (competent) activity. Functional analyses further demonstrated that either PEST-mutant VENTX (undegradable) gain-of-function (GOF) or MEK1 loss-of-function (LOF) result in symmetric distribution of VENTX in mitotic PSCs and prolonged maintenance of refractory and undifferentiated state in vivo [14]. Thus, MEK1-mediated asymmetric cell distribution of VENTX and the resulting PSCs heterogeneity is mandatory for the pluripotency *continuum* in vivo (Figure 2B,C), as well as SCF-β-TRCP ubiquitin-mediate global clearance of VENTX in committing embryonic cells at the onset of morphogenesis (i.e., gastrulation) in vivo [14,62]. 

Studies in hESCs strengthen the role of asymmetric cell division (ACD) in conferring diverging fates to mitotic pluripotent stem cells (PSCs), whereby one daughter cell maintains refractory/naive state (High-NANOG) whilst the other daughter becomes competent (Low-NANOG) to respond to pro-differentiation cues [63,64]. This would explain why inhibition of MEK1 activity, or undegradable forms of VENTX/NANOG, can lock vertebrate PSCs in a naïve/refractory pluripotent state [1,14,24,59,60,61]. Furthermore, PSCs ACD suggests that the decision to commit is largely determined before the pro-differentiation cue is transcriptionally effective and can be predicted by a cell’s pre-existing VENTX/NANOG protein distribution (Figure 2C).

Whereas VENTX/NANOG acts as a “locker” of the pluripotent refractory/naive state in PSCs, several studies point out that POU5/OCT4 functions in a bi-modal manner: POU5/OCT4 can both interact with VENTX/NANOG in maintaining refractory pluripotency [16,26,33], as well as with multiple epigenetic (e.g., JARID2, CBX1, SMARCA4) and transcriptional (ZIC, OTX, SOX-B1, SOX-F) competence factors (Figure 2A) [16,26,33,65,66,67]. Such a competing POU5/OCT4 interaction network is crucial for the biological function of POU5/OCT4 for pluripotency *continuum*, and studies in vivo strengthen the relevance of VENTX/NANOG in competing and modulating the transcriptional activity of POU5/OCT4-SOX-F complex, as well as SMADs and CTNNB1, in early cell fate commitment [16,26,27,28,33,34,35,36]. In parallel, other competence factors (e.g., JARID2, CBX1, SMARCA4, BAP1, KDM5A, KMT5B, SNAI1, SOX9) participate in the transition from refractory/naive to competent/formative pluripotency and further multi-lineage commitment, as demonstrated in *Xenopus* and ESCs [65,66,67,68,69,70,71,72]. Importantly, POU5/OCT4 can prepare PSCs to transit from refractory to competent pluripotency by stimulating MEK1 activity through *FGFs* ligand expression [56,57,73]. In turn, MEK1 could destabilize VENTX/NANOG, thus favoring POU5/OCT4 competitive interaction with competence factors. Thus, the crosstalk between POU5/OCT4 and FGF/MAPK may create a window of opportunity for PSCs to escape refractory state imposed by VENTX/NANOG and to realize the multi-lineage potential by the acquisition of the competence to interpret as and to respond to pro-differentiating cues (e.g., WNTs, FGFs, TGFβ) (Figure 2A). In vertebrate blastula/epiblast, the spatial distribution of pro-differentiating signals (e.g., WNTs, TGFβ) determine the intensity and duration of the activity of down-stream effectors (i.e., CTNNB1-TCF/TLE, SMADs) and chromatin organization in the PSCs transition from refractory to competent pluripotency, thus establishing cell fate decision and germ layer commitment at the onset of gastrulation [7,8,74,75,76,77,78,79,80,81,82,83,84]. Direct visualization of active signaling in vivo showed that the intensity of these signals increases during PSCs transition from refractory to competent state, and that variations of the resulting morphogenetic gradient is predictive of the spatial cell commitment within the forming germ layers [85,86,87,88]. Studies in zebrafish, *Xenopus*, chicken, and mammalian embryos, as well as in ESCs in vitro, shed light on the combinatorial logic behind the activity of pro-differentiating signals (e.g., WNTs, TGFβ) to activate the transcriptional program(s) involved in lineage commitment: high activity of endogenous WNT/CTNNB1 and TGFβ/SMADs drives mesendoderm commitment, whereas low activity of endogenous WNT/CTNNB1 and TGFβ/SMADs allows ectoderm commitment [77,78,79,80,81,82,83,84] (Figure 2A). FGF/MAPK signaling activity is required for the induction of both ectoderm and mesendoderm lineages, as well as for further lineage-restricted decisions (e.g., neural and neural crest cell fate; axial/paraxial mesodermal fate) [81,89,90,91]. Recently, several studies point out the fundamental function of chromatin organization and poised enhancers, which represent distal regulatory elements that control the expression of major developmental genes, during the process of embryonic cell commitment and the establishment of stable epigenetic memory, which signs the exit from pluripotency state at the onset of gastrulation [68,69,70,92,93,94].

Altogether, these data suggest that PSCs transition from refractory/naïve to competent/formative state does not occurs in a synchronous and homogeneous manner, but rather PSCs represent an heterogenous population with biased competences to execute cell commitment depending on the levels of VENTX/NANOG protein and their physical interaction with POU5/OCT4. The significance of the asynchronous exit from refractory/naïve pluripotency and the heterogeneous competence of PSCs to commit need more attention in order to gain better insight on embryonic cell diversification and commitment.

## 3. Reprogramming Capacity of Developmental Potential Guardians Shapes Neural Crest Multipotency and Vertebrate Evolution

The neural crest (NC) is a vertebrate-specific cell population that is specified at the neuro-epithelial border (NEB) and that is empowered with a broader developmental potential than the ectodermal lineage of origin (Figure 3). Together with the three primary germ layers (i.e., ectoderm, mesoderm, endoderm), NC is often referred as the “fourth germ layer” due to its multipotency [95]. The NEB cells are located into the ectodermal layer and are specified by inductive signals (e.g., FGFs, WNTs) from the overlaying mesoderm at gastrulation. The neuro-epithelial border cells (NEBs) form a transient embryonic cell population. Investigations on the transcriptional state of vertebrate NEBs by scRNA-seq analyses in several vertebrate models (i.e., zebrafish, *Xenopus*, chicken) have shown that NEBs express the NEB-specific gene regulatory network (GRN: *PAX3*/*7*, *ZIC1*, *MSX1*/*2*, *TFAP2*, *ZNF703*) (Figure 3A) together with ectodermal-lineage specifiers (*SOX2*/*3*, *GMNN*, *LHX5*) [37,96,97]. These studies thus propose that NEBs initially transit through an ectodermal state. Consistent with this idea, *Xenopus* NEBs spontaneously differentiate into sensory neurons when isolated or transplanted [98], whereas mammalian (i.e., mouse and human) NEB stem cells are transcriptionally biased toward a neuroectodermal state (*PAX3*, *ZIC*, *MSX1*, *SOX2* +) and spontaneously differentiate toward sensory neurons (DCX, POU4F1 +) in vitro [80].

At the end of gastrulation, vertebrate NEBs segregate in two distinct cell populations: (1) neurogenic placodal ectoderm, an unipotent neurogenic cell population of the anterior ectoderm that express *SIX1* and *EYA1* and give rise to sensory organs in the head, and (2) multipotent neural crest cells (NCCs), which express NC-specific GRN *SNAI1*/*2*, *FOXD3*, *SOX8*/*9*/*10* (SOX-E family) (Figure 3A) [95,99]. The transition from NEB-to-NC is dependent on WNT signaling pathway, as observed in vertebrate embryos and mammalian NEBs [77,78,79,80,81,82,100]

During neurulation, NCCs loss neuroepithelial features through an epithelial-mesenchymal transition (EMT) process, delaminate from the closing neural tube, migrate and then differentiate into a large spectrum of differentiated cell types: craniofacial bones, cartilages, muscles and the heart outflow mesenchyme, thyroid cells, secretory cells of the adrenal medulla, peripheral nervous system (PNS) and melanocytes (Figure 3A) [95,99].

The presence of the “new head” rudiments in fossil records (e.g., *Myllokunmingia*, *Haikouella*, *Haikouichthys*) suggest that the NCCs (and the vertebrate ancestor) arose during the Cambrian period [95]. Whereas NCCs are specified along the anterior–posterior axis, pioneering studies on extant amniote models proposed that cranial NCCs were unique in forming ectomesenchyme derivatives [95,99]. However, according to recent paleontological and embryological analyses in alternative vertebrate models (e.g., *Leucoraja erinacea*/little skate; *Petromyzon marinus*/sea lamprey), the broader developmental potential of NCCs was distributed along the anterior–posterior axis, as deduced from the exoskeletal armor of Ostracoderms (e.g., *Hemicyclaspis*; Agnatha) and Placoderms (e.g., *Dunkleosteus;* Gnathostomata) fossil record, trunk NC-derived ectomesenchyme dermal denticle in cartilaginous fish, and latent ectomesenchyme potential of amniote trunk NCCs in vitro [95,99,101,102,103,104]. Strikingly, the sum of data from invertebrate chordates (e.g., Urochordates, *Ciona*/sea squirt; Cephalochordates, *Branchiostoma*/amphioxus) and protostomes (e.g., *Caenorhabditis*/nematod; *Platynereis*/annelid) showed that the invertebrate neuro-epithelial border possesses neurogenic potential and lack multipotency but share NEB/NC-GRN (i.e., *pax3*/*7*, *zic*, *msx*, *tfap2*, *znf703*, *snai*, *foxd*, *sox8*/*9*/*10*) with vertebrate NEB/NC [50,51,52,99,105,106,107,108,109,110,111] (Figure 3B). Hence until recently, the molecular mechanism(s) empowering vertebrate NEB with the competence to give rise to multipotent NC remained obscure. Phylogenetic analyses demonstrated that both *VENTX*/*NANOG* and *POU5*/*OCT4* are vertebrate-specific genetic innovations [15,23] activated in vertebrate NEBs at the onset of NCCs specification [15,45,46,47,48]. Functional analyses in *Xenopus* demonstrated that *ventx2*, belonging to the *VENTX*/*NANOG* family, drives the “endogenous reprogramming” of vertebrate NEBs to NCCs in vertebrate embryo [15] (Figure 3C). Mechanistically, VENTX/NANOG regulate expression of stem cell markers (*POU5*/*OCT4*, *TERT*) and epigenetic memory erasers (*TET3*, *KDM4A*, *SMARCA4* and *CHD7*) in order to promote reprogramming of NEBs-to-NCCs [15]. Accordingly, NCCs-specific ventx2 LOF specifically abrogates NCCs multipotency, ectomesenchyme potential and craniofacial development, without affecting the neurogenic program in NEBs, which is the shared cell lineage between vertebrate NEBs/NCCs and bilaterian NEBs [106,107,108,109]. 

The loss of ectomesenchyme potential and craniofacial development after ventx2 LOF can be rescued by mammalian Nanog. Furthermore, ventx2 is sufficient and necessary to reprogram, together with NEB specifiers (i.e., *PAX3*, *ZIC1*), refractory differentiated epithelial ectodermal cells to immature and undifferentiated NCCs in vivo and in vitro [15]. Thus, when compared to the unipotent NEBs of invertebrates, VENTX/NANOG would empower the ectodermal NEBs of the proto-vertebrate ancestor with a new and broader competence to acquire alternative cell fates (ectomesenchyme) but permissive to the ectodermal (neurogenic) potential, thereby promoting the rise of multipotent NEBs/ NCCs, the “new head” and the vertebrate subphylum. [15] (Figure 3D). Gradual acquisition of new and axial-specific regulatory sub-circuits further allowed evolution of gnathostome NCCs derivatives [104,112], in which reprogramming processes still link at the heart of the broad developmental potential of NNCs and their competence to acquire alternative cell fates during evolution [15,112,113,114]. Consistent with these findings, POU5/OCT4 acts similarly to VENTX/NANOG in mouse developing NCCs [46]. Therefore, the rise of *vsDPGs* and multipotent NCCs should be considered a major step in the rise and evolution of vertebrates, as loss of *vsDPGs* in vertebrate multipotent NEBs can be interpreted as a regression to a primitive/atavistic unipotent condition that is functionally comparable to invertebrate NEBs. 

They are mechanistically linked to endogenous in vivo reprogramming of neuro-epithelial border cells towards the neural crest cells potential [15,46]. As a result, the epigenetic memory of vertebrate ancestor NEBs/NCCs became permissive for the activation of a pre-existing ectomesenchyme genetic program in parallel with ectoderm genetic program, whilst the epigenetic memory of invertebrate NEBs maintains a bias toward neurogenic program and ectomesenchyme remains exclusive of mesendoderm derivatives [105,106,107,108,109]. These data help to explain the origin and the biological relevance of vsDPGs as reprogramming factors [18,19,29,30,31,32], which could arise, be selected and shaped by their function in NCCs. Further addition of new genetic sub-circuits during vertebrate evolution allowed the rise of new and alternative cell types from NCCs [104,112,113,114,115,116] with a direct impact on the survival, adaptability and fitness of the organisms. Interestingly, other factors involved in control of early pluripotency are reactivated in NCCs (e.g., *lin28A*) that regulate NCCs multipotency downstream to the WNT signaling pathway, which is secreted from the dorsal neural tube and acts as “positional information” cue [100].

Due to their exploratory behavior and multipotency, NCCs have been recognized as a key contributor to phenotypic plasticity and evolvability [11,12,15,117,118,119,120,121,122]. It will be interesting to understand whether changes in *VENTX*/*NANOG* and *POU5*/*OCT4* spatiotemporal expression or protein activity (e.g., stability/degradation, physical interactions with co-factors and/or DNA) and distribution during mitoses (asymmetric cell division) within NCCs may contribute to cellular heterogeneity and fate choice during development, as observed in PSCs [15].

Changes in vsDPGs expression and activity may be responsible for the phenotypic variation of NCCs derivatives via differential cell/tissue proliferation, migration, timing of differentiation and, ultimately, tissue growth and shape [117,118,119,120,121,122,123,124]. New tools allowing precise spatiotemporal modulation of gene activity at cellular resolution may allow to tackle these questions [125,126,127,128,129,130].

## 4. Developmental Potential Guardians Role in Neuro-Mesodermal Progenitors and Vertebrate Axial Length

As demonstrated by in situ hybridization (ISH) and scRNA-seq analyses in *Xenopus* and zebrafish embryos, vertebrate-specific developmental potential guardians (*vsDPGs*) are expressed in the posterior neuro-mesodermal progenitors (NMPs), an undifferentiated and multipotent cell population that participate in neural tube, trunk NC, somites and notochord formation during posterior axis elongation [16,22,39,48,131] (Figure 1C). A conserved feature of vertebrate NMPs is the co-expression of lineage-specific mesodermal (*T*/*BRA*) and neural (*SOX2*) regulators that, *in concerto* with differentiating cues (e.g., FGFs, WNTs, Retinoic Acid/RA, TGFβ) coordinate NMPs differentiation into neural/NC or mesodermal lineages [39,74,75,76,131,132,133]. Functional analyses in *Xenopus* and zebrafish suggested that vsDPGs may participate in the maintenance of undifferentiated and quiescent NMPs since vsDPGs LOF lead to posterior axis truncation [17,21,22,134]. Similarly, mammal and chicken NMPs contribute to both neural, neural crest and mesodermal posterior derivatives and co-express *T/BRA* and *SOX2* [132,135,136]. Whereas zebrafish NMPs are multipotent up to bud stage/early somitogenesis and then a quiescent population of NMPs is maintained up to the end of somitogenesis [131,133,137], amniotes NMPs are maintained multipotent throughout somitogenesis [132,135,136] and this is likely due to the difference in proliferation and volumetric growth and proliferative phase(s) observed among amniotes versus non-amniotes [131]. Thus, vertebrate NMPs are competent to generate both neural/NC or mesodermal cell fates, and this potential is linked to the global proliferation rate, the stage of development, volumetric growth and, ultimately, the mode of development (namely, fast developmental rate in amniotes *versus* slow developmental rate in non-amniotes) [131,133,137].

Conditional manipulation of vsDPGs activity in gastrulating mouse embryo impact on posterior axis development through *HOX* expression [138,139]. Strikingly, modulation of *POU5*/*OCT4* activity in mouse trunk NMPs impact on the growth and length of the posterior axis and induces a more posterior shift of *HOX* gene-expression boundaries in the extended trunk [138]. This lead to hypothesize that exacerbated posterior axis length observed in snakes (or eels) requires a sustained and prolonged maintenance of NMPs pool by vsDPGs in order to ensure a cellular source for forming neural tube, PNS, notochord and somites.

Therefore, it is tempting to speculate that vsDPGs might confer variability and evolvability to the organism by modulating the harmonious growth and shape of the posterior axis, as observed in a snake’s body length. Since the timing of NMPs fate-restriction, cell rearrangement and volumetric growth of the posterior body axis varies among vertebrate species [131,133] the contribution of vsDPGs to the posterior axis development in vertebrates and their evolution, a consequence of their function in early pluripotency, need more accurate analysis in vivo. Comparative molecular-cellular analyses in closely related species, together with genomic data and mathematic/morphometric modelling, may ultimately tackle the question about the mechanism(s) by which evolution operate [11,12,115,116,131,132,133,135,136,137,140,141,142,143,144].

Altogether, the roles of vsDPGs in NCCs/Head and NMPs/Trunk-Tail suggest an exciting and yet poorly explored contribution of vsDPGs to the global architecture of a living organism acting through tissue growth/shape to phenotype and evolution. Several studies have investigated the mechanism(s) at the heart of vertebrate phenotypical/morphological variability and evolution [117,118,119,120,121,122,131]. It will be interesting to understand whether the evolutionary dynamics of vsDPGs (i.e., duplications and loss), the variations in functional sites modulating their stability and their physical interactions [65,66], as well as modifications occurring in DNA sequences (e.g., promoters, enhancers) and the epigenetic modifications controlling their expression may help to infer causal mechanism(s) driving phenotypical variations among species. Since *VENTX*/*NANOG* and *POU5*/*OCT4* evolutionary history show an intriguingly high degree of complexity [14,15,22,23] (Figure 1B), it will be important to understand whether their evolutive dynamics correlates with vertebrate phenotypical variations among species.

## 5. Developmental Potential Guardians in Human Diseases

Due to its growth, exploratory and invasive behavior, resistance to therapies, and regenerative capacity/relapse, cancer can be considered to be an “*alien organism*” that exploits and parasites the living “host” from which it originates. Several mutation-driver genes (Mut-driver-genes) have been characterized in cancer cells so far (e.g., *KRAS*, *BRAF*, *MYC*, *TP53*), however vsDPGs have gained attention in the process of carcinogenesis due to their tumorigenic potential [145,146,147]. *VENTX* and *NANOG* were found highly expressed in brain (glioma/glioblastoma) [148,149,150], pancreatic [151,152,153] renal [154,155], esophageal [156,157] and testicular cancers [28,158,159]. Interestingly, VENTX and NANOG share activity in hematopoiesis by repressing the genes responsible for terminal differentiation (e.g., *TAL1*, *KLF1*) [160,161], as well as promoting leukemia [161,162,163]. *VENTX* and *NANOG* are highly expressed in CD34^+^ leukemic stem cells (LSC, a subpopulation responsible for drug resistance, metastasis, and leukemia relapse) and their depletion blocks AML proliferation and growth [161,162,163].

Whereas *NANOG* function has been extensively characterized in cancers, mainly for its expression and activity in cancer stem cells (CSCs), less is known about *VENTX*. Thus, the BioGRID Open Repository of CRISPR screens (BioGRID-ORCS database) [164] shows that *VENTX* is involved in growth/proliferation/resistance of several cancer cell lines (e.g., brain, pancreatic, renal and ovarian cancers) (Table 1). Intriguingly, *VENTX* is important for the proliferation of neural stem cells (NSCs), Glioma, and Glioblastoma (Table 1), thus suggesting an important role for *VENTX* both in normal/physiological and abnormal conditions of the human brain. Accordingly, Gene Expression Profiling Interactive Analysis (GEPIA2 database) based on primary tumors and normal samples from the TCGA and the GTEx databases [165] shows that *VENTX* is highly expressed in both low-grade-glioma (LGG) and glioblastoma (GBM) when compared to normal samples (Figure 4A) and scRNA-seq data (available on Broad Institute Single Cell Portal database) further shows that *VENTX* is expressed in malignant GBM cancer cells (Figure 4B) [166].

Survival rate based on *VENTX* expression levels (Figure 4C) suggests that *VENTX* may be used in the prognostic of brain cancer development and patient survival as well. Thus, it would be relevant to better understand the function of *VENTX* in human brain physiology and carcinogenesis.

*POU5*/*OCT4* can initiate reprogramming and carcinogenesis in vivo [167]. Functional analyses strengthen the biological and molecular relevance of *POU5F1*/*OCT4* in controlling carcinogenesis and cancer stem cells (CSCs) [168,169,170,171], pointing to yet obscure cellular and molecular features likely shared between carcinogenesis and reprogramming. Intriguingly, vsDPGs are expressed in developing human primordial germ cells (PGCs), in the germ line and are strongly up-regulated in germ cell tumors (i.e., seminoma and non-seminoma) [28,40,41,42,43,44,123,159]. Since PGCs arise from an “endogenous reprogramming” of the epigenetic memory of mesendoderm progenitors [172,173,174], similar to what observed in embryonic NEBs-to-NCCs [15,46,113], it is tempting to speculate that some process operating in the vertebrate embryo, and under the control of vsDPGs (i.e., refractory pluripotency maintenance, NCCs, and PGCs reprogramming), may be abnormally reactivated in normal cells undergoing malignant transformation to cancer cell of origin.

Altogether, vsDPGs play a role in carcinogenesis and therapeutic resistance, suggesting that they may present crucial targets to counteract cancer development, aggressiveness and relapse. It is intriguingly to note that vsDPGs are not Mut-driver-genes but, due to their aberrant expression levels, mainly represent Epigenetic-driver-genes (Epi-driver-genes) [145]. How and when vsDPGs are reactivated in cancer cells is still poorly understood, as well as their relevance in establishing and maintaining cancer heterogeneity. Answering these questions may shed new light on the hierarchy of events leading to carcinogenesis and lead to more precise prophylactic and therapeutic approaches [147]. Due to their tumorigenic and reprogramming potential, it will be crucial to understand whether vsDPGs exclusively participate in cancer progression and resistance, or if they physiologically act earlier than expected during malignant transformation of a normal cell to a cancer cell of origin in vivo [175], thus prior to the appearance of a tumor mass and intertumoral cell heterogeneity. This should improve strategies for future preventive therapeutic approaches targeting cancer cell(s) at their early stages.

## 6. Conclusions and Perspectives

Vertebrate-specific developmental potential guardians (vsDPGs) have allowed us to delineate the molecular and cellular bases of the embryonic developmental potential in vivo. VsDPGs control the pluripotency *continuum* and competence for multilineage commitment in vertebrate pluripotent stem cells (PSCs), as well as later reacquisition of multipotency in neural crest cells (NCCs) through an “endogenous reprogramming” process. Furthermore, vsDPGs activity in vertebrate neuro-mesodermal progenitors (NMPs) participates in posterior axis growth and elongation. Altogether, these findings suggest that vsDPGs are key players of embryogenesis, but also suggest a less explored function in vertebrate evolution.

Since cell types expressing vsDPGs have been described as heterogeneous populations, it would be interesting to understand whether inter-specific variations in vsDPGs activity may impact on the growth, morphology and shape of the organism. In fact, most of these studies display technical limitations since the function of vsDPGs has been analyzed with limited spatial and/or temporal control. As an example, conditional vsDPGs GOF and/or LOF have been done in the whole organisms or cell types, with catastrophic effect on the organism or tissue(s). This may be useful for developmental and morphological analyses, but clearly this approach prevents an understanding of vsDPGs role at single cell level. For example, specific vsDPGs LOF in vertebrate NCCs allows one to elucidate the global endogenous reprogramming process conferring multipotency to vertebrate NEBs/NCCs. However fine spatial-temporal variations in vsDPGs activity in single NCCs should avoid the global catastrophic effects previously described and may be useful in elucidating their role in NCCs stemness and in shedding light on how the dynamics of undifferentiated state may impact on cell fate, tissue growth, and morphogenesis. It is interesting to note that WNT signaling pathway controls the early undifferentiated state of pre-migratory NCCs in chicken, as well as morphological and phenotypical variation of skull in fishes [100,118]. It has been thus suggested that variations in the activity (and downstream targets) of signaling pathways (e.g., WNTs, BMPs, SHH) may contribute in vertebrate morphological and phenotypical variation, adaptation and evolution [117,118,119,120,121,122]. Since vsDPGs can physically interact with downstream effectors (CTNNB1, SMADs) of such pathways (e.g., WNTs, BMPs) and are expressed in embryonic cells (i.e., NCCs and NMPs) contributing to morphological and phenotypical variation among vertebrates, it is reasonable to assume that variations in vsDPGs activity may contribute to vertebrate phenotypical variations and evolution, beyond their role in PSCs. Therefore, the development of new approaches allowing precise spatiotemporal single cell manipulation through LOF/GOF of target genes, such as vsDPGs, may help to understand the dynamics of tissue formation, morphogenesis, stemness during ontogenesis, and to infer phenotypical changes throughout evolution.

Recently, versatile optogenetic approaches have been developed to control gene expression and protein activity in a live animal at single cell level and with temporal resolution of a few seconds [129,176]. Among them, an original and fast optogenetic approach is based on a conditional ERT/caged Cyclofen-OH (cCYC) induction system in vivo, allowing the activation of specific genes either permanently (by using Cre-ERT/loxP system) or transiently (by using a Gal4-ERT/UAS system) [125,126,127,128,129,130]. Such an optogenetic approach allows precise spatiotemporal control of gene expression and protein activity at single cell level or in few cells. This strategy is compatible with the photoactivation of a wide variety of proteins. Therefore, optical methods open opportunities for the local spatiotemporal investigation of developmental processes, identification (and manipulation) of stem cells, and the study of carcinogenesis at single cell level in a live organism [127,176]. This strategy may provide crucial information about vsDPGs activity in development and differentiation/reprogramming in vivo with unprecedented spatiotemporal resolution.

Can vsDPGs affect the evolvability of the system acting on the unit of evolution (i.e., the cell) and how? Studies in vertebrates suggested that vsDPGs can confer cell plasticity/adaptability during development. Hence, precise manipulation of the spatiotemporal activity of vsDPGs at single cell level by optogenetic approaches may ultimately answer the question about the rise and changes of shapes in evolution. Quantitative and biophysical approaches at multi-scale resolution (from single cell to tissue/organ), together with mathematical modelling and imaging, can ultimately help to elucidate how the fundamental and common laws governing morphogenesis during ontogeny and phenotypical variations throughout evolution intersect with vsDPGs activity in cells [11,12,115,116,124,131,132,133,135,136,140,141,142,143,144].

Furthermore, manipulating vsDPGs at single cell level may help to better understand the process of carcinogenesis, the malignant transformation of a normal cell to a cancer cell of origin, how heterogeneity arise among cancer cells and, ultimately, be useful to develop new and patient specific therapeutic approaches targeting each cell type via specific anti-cancer agents [145,146,147,175]. Therefore, the development of new tools allowing for genetic modifications (e.g., gene editing) with high spatiotemporal resolution would have a great impact on the whole scientific community and should revolutionize our knowledge of the rules governing animal development, morphogenesis, shape and evolution, as well as the origin of human diseases.

## Figures and Tables

**Figure 1 cells-11-02299-f001:**
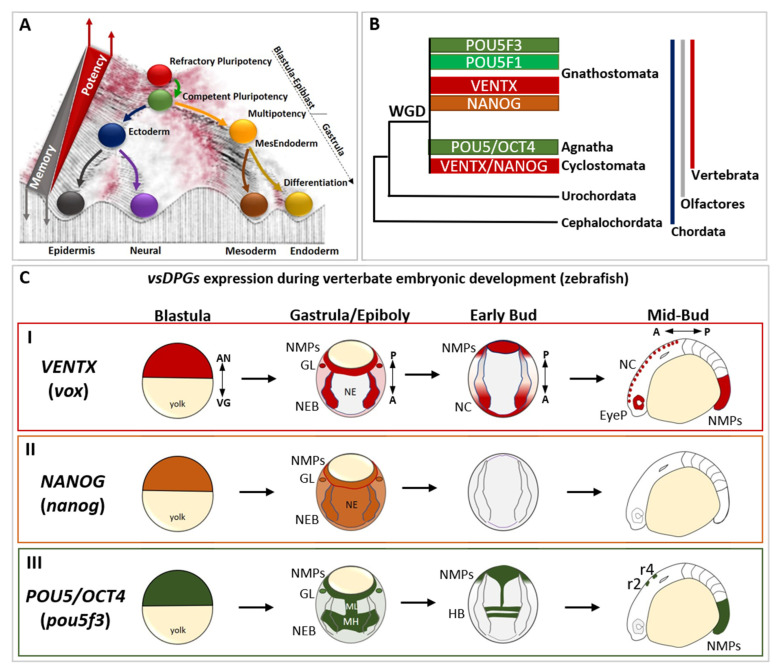
The evolutionary history and the developmental expression of vertebrate-specific developmental potential guardians (vsDPGs) *VENTX*/*NANOG* and *POU5*/*OCT4.* (**A**) Modified Waddington’s epigenetic landscape illustrating the cell’s developmental potential changes during development. Cells are represented as balls rolling down a valley, which metaphorically represent the embryonic development *continuum*. At the top of the hill there are refractory pluripotent stem cells (PSCs; red ball), which represent blastula/epiblast cells in vivo. Once PSCs became competent to respond to differentiating cues (green ball), embryonic cells exit pluripotency and commit into embryonic cells of primary germ layers (i.e., ectoderm/blue ball, MesEndoderm/orange ball). In this step, which occurs at the onset of gastrulation, embryonic cells become multipotent and shape their transcriptional, cellular/molecular and epigenetic memory (grey triangle). During morphogenesis, embryonic cells differentiate into specific cell types (neural/purple; epidermis/black; mesoderm derivatives/dark yellow; endoderm derivatives/golden yellow), loose their multipotency and establish their epigenetic memory (represented as a barrier separating balls/cells in the landscape). Note that the red part of the landscape, as well as the top of the barrier, represent the points where cells possess higher developmental potential, whereas the dark parts of the landscape represent the points where cells possess lower developmental potential. This suggests that, at any given time of development (or in an adult) cells might “jump” to the top of the barriers to regain high developmental potential (as in regeneration, dedifferentiation) or go back to a pluripotent state (as in reprogramming process). (**B**) Simplified cladogram representing the evolutionary history of vsDPGs *VENTX*/*NANOG* (in red and orange) and *POU5*/*OCT4* (in green) during the evolution *of* the phylum Chordata (blue line), the clade Olfactores (grey line) and the sub-phylum Vertebrata (red line). vsDPGs are absent in the genus of Cephalochordata (e.g., *Branchiostoma*/amphioxus) and Urochordates (e.g., *Ciona*/sea squirt) sub-phyla. Furthermore, vsDPGs have been found in the genomes of extant vertebrates, both in species belonging to the infraphylum Agnatha (e.g., *Cyclostomata*/lampreys and hagfishes) and species belonging to the infraphylum Gnathostomata (e.g., *Chondrichthyes*/sharks and rays; *Osteichthyes*/*Actinopterygii* such as Teleostei *plus Sarcopterygii* such as Tetrapods). Note that it has been proposed that vsDPGs likely arose as a result of whole genome duplication (WGD) experienced by the last common ancestor of extant vertebrates. Modified from Scerbo P et al., 2020 [15]. (**C**) Expression of vsDPGs during vertebrate embryonic development (zebrafish) (I) The zebrafish ortholog of human *VENTX* (known as *vox*, in red) is expressed in pluripotent blastula cells, in neuro-epithelial border cells (NEBs), germ line (GL), and posterior neuro-mesodermal progenitors (NMPs) during gastrula/epiboly stage, but absent in developing neuroectoderm (NE). At early-bud stage (onset of somitogenesis), *vox* expression is maintained in neural crest (NC) and NMPs, whereas at mid-bud stage (5–9 somite stage) *vox* is expressed in NC, developing eye primordium (EyeP) and NMPs. The zebrafish ortholog of human *NANOG* (*nanog*, in orange) is expressed in pluripotent blastula cells, and ubiquitously during gastrula/epiboly stage. At bud stages (somitogenesis) no expression has been detected. (III) The zebrafish ortholog of human *POU5/OCT4* (known as *pou5f3*, in green) is expressed in pluripotent blastula cells, in neuro-epithelial border cells (NEBs), germ line (GL), and posterior neuro-mesodermal progenitors (NMPs) during gastrula/epiboly stage, with strong expression in presumptive midbrain-hindbrain (MH) precursors and neuroectodermal mid-line (ML). At early-bud stage (onset of somitogenesis), *pou5f3* expression is localized in hindbrain (HB) precursors and NMPs, whereas at mid-bud stage (5–9 somite stage) *pou5f3* marks rhombomeres 2 (r2) and 4 (r4) in the hindbrain and posterior NMPs. The embryonic axes are indicated as: animal pole (AN), vegetal pole (VG), A (anterior), P (posterior). The expression profile of vsDPGs has been schematized by using in situ hybridization (ISH) and scRNA-Seq data for *vox*, *nanog* and *pou5f3* on (https://zfin.org at 1 July 2022) and (https://kleintools.hms.harvard.edu/paper_websites/wagner_zebrafish_timecourse2018 at 1 July 2022).

**Figure 2 cells-11-02299-f002:**
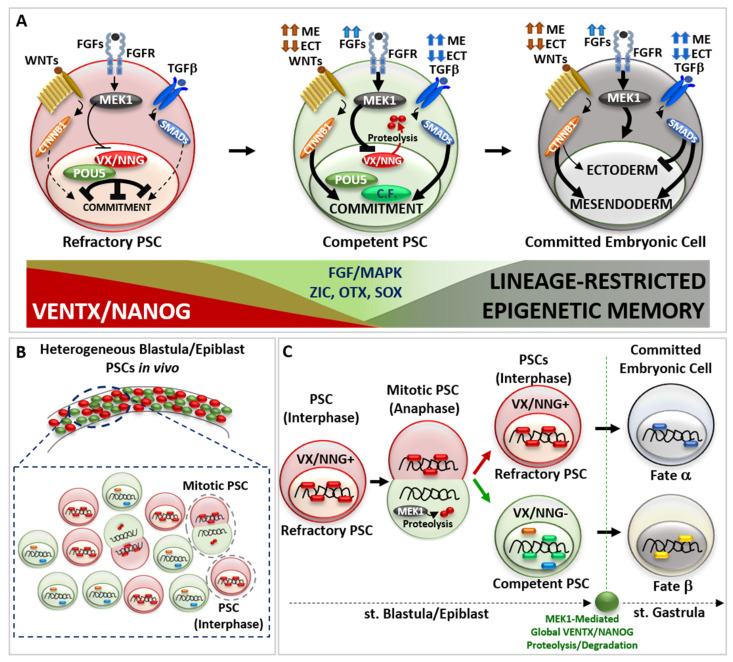
The *continuum* of cell potency in vivo. (**A**) Schematic representation of pluripotent stem cell (PSC) transition from a refractory (red cell) to a competent (green cell) state and further germ-layer specific embryonic cell commitment (dark cell). Refractory PSC displays high activity of VENTX/NANOG—POU5/OCT4 heterodimer, which counteract the activity of pro-differentiating cues (extracellular ligand TGFβ, WNT signalling pathways) and cell commitment. Once increasing MEK1 kinase, acting down-stream of the FGFs signal (indicated by blue arrows), destabilise VENTX/NANOG—POU5/OCT4 heterodimer through VENTX/NANOG proteolysis, competent PSCs can interpret and respond to pro-differentiating cues (TGFβ, WNT), thus entering into commitment through new interactions between POU5/OCT4, Competence factors (C.F., light green) and pro-differentiating intracellular effectors (CTNNB1, SMADs). Note that the high or low intensity of pro-differentiating signal (TGFβ/SMADs indicated by dark blue arrows, WNT/CTNNB1 indicated by brown arrows) instruct competent PSCs to commit to a germ layer specific cell fate, which is realized once PSCs exit from pluripotency and enter into embryonic cell commitment (dark cell). On the bottom: schematic representation of VENTX/NANOG activity in PSCs (in red, high in refractory PSCs, low in competent PSCs) compared to FGF/MAPK and Competence factors (i.e., ZIC, OTX, SOX) (in green, low in refractory PSCs, high in competent PSCs) and the appearance of lineage-restricted epigenetic memory (in dark, absent in refractory PSCs and increased in competent-to-committed embryonic cells). (**B**) Schematic representation of refractory to competent PSC transition in vivo. In vertebrate blastula/epiblast, mitotic PSCs divide asymmetrically and generates one refractory daughter PSCs (red cell) and one competent daughter PSC (green cell). (**C**) Mitotic blastula/epiblast PSC show asymmetric distribution of VENTX/NANOG (VX/NNG) protein at anaphase. MEK1 mediates VENTX/NANOG degradation in one daughter PSC. At the end of mitosis, the refractory daughter PSCs (red cell) inherits VENTX/NANOG protein (VX/NNG+), whereas the competent daughter PSC (green cell) does not inherit VENTX/NANOG (VX/NNG-) and becomes competent to respond to differentiating cues (green, blue and orange circles). At the onset of gastrulation, global clearance of VENTX/NANOG protein occurs in embryonic cells in a MEK1-dependent manner. The described mode of asymmetric cell division (ACD) is important for pluripotency continuum in vivo and will impact on the cell fate (hypothetical fate α versus β) of embryonic cells during gastrulation.

**Figure 3 cells-11-02299-f003:**
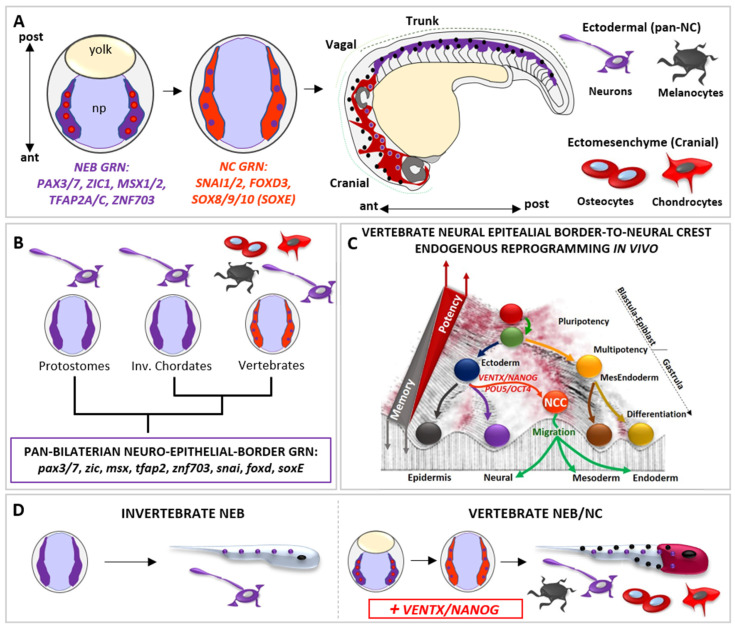
The vertebrate-specific neural crest cells (NCCs) are empowered with multipotency and ectomesenchyme potential by vsDPGs *VENTX*/*NANOG* and *POU5*/*OCT4*. (**A**) Schematic representation of neural epithelial border cells (NEBs, in purple *plus* red dots, gastrula stage) specification to neural crest cells (NCCs, in light red *plus* purple dots, neurula stage) during vertebrate development. Vertebrate NEBs are specified during gastrulation and activate the NEB-specific gene regulatory network (GRN) centred on *pax3*/*7*, *zic1*, *msx1*/*2*, *tfap2* and *znf703* orthologs (in purple). At the onset of neurulation, NEB-GRN activate the down-stream NCC-GRN, centred on *snai1*/*2*, *foxd3*, *sox8*/*9*/*10* (*soxE*) orthologs (in light red). During morphogenesis, migratory NCCs (Cranial NCCs in dark red, Trunk NCCs in purple) migrate from the dorsal neural tube along the anterior-posterior (A/P) axis, loose multipotency and become competent to respond to pro-differentiating cues. Committed NCCs colonize new embryonic loci and differentiate into ectomesenchyme cranial derivatives (e.g., chondrocytes and osteocytes of the skull, cells represented in dark red) and ectodermal derivatives (e.g., neurons and melanocytes along the A/P axis, cells are represented by purple and black dots). (**B**) Simplified cladogram representing bilaterian phyla where NEBs (in purple; protostomes and invertebrate chordates) are unipotent and neurogenic (purple; neuron) and where NEBs give rise to multipotent NCCs (in light red plus violet dots, vertebrates). Note that, based on comparative and functional studies, the GRN specifying both invertebrate NEBs and vertebrate NEBs/NCCs show a high degree of conservation (referred as PAN-BILATERIAN NEURO-EPITHELIAL BORDER GRN) and is thus constrained throughout bilaterian evolution. This implies that the well characterized vertebrate NCCs-GRN is ontogenetically specific to NCCs but not phylogenetically specific. (**C**) The Waddington’s epigenetic landscape illustrates the cell’s developmental potential changes during development and the regain of potency of NEBs-NCCs by the reactivation of vsDPGs *VENTX*/*NANOG* and *POU5*/*OCT4*. Note that such a regain of multipotency in NEBs-NCCs is due to an endogenous in vivo reprogramming process promoted by vsDPGs, which impact on the epigenetic memory of ectodermal cells transitioning to multipotent NEBs/NCCs. Since barriers represent the epigenetic memory in the Waddington’s landscape, it is important to note that NCC (in light red) has not been placed at the top of the valley (as pluripotent stem cells, PSCs) but on the top of the barrier of a given developmental time (end of gastrulation). (**D**) Schematic illustration of the neural epithelial border (NEB) development and terminal differentiation in invertebrates (left) and in vertebrates (right). The vertebrate multipotent NEB/NC (in light red *plus* purple) evolved from an ancestral unipotent condition (in purple) shared with invertebrates, thereby the introduction of VENTX/NANOG activity conferred multipotency and acquisition of ectomesenchyme potential (represented by terminally differentiated chondrocytes and osteocytes of the skull, in red) together with the ectodermal potential (represented by terminally differentiated neurons and melanocytes along the A/P axis, represented by purple and black dots along the A/P axis).

**Figure 4 cells-11-02299-f004:**
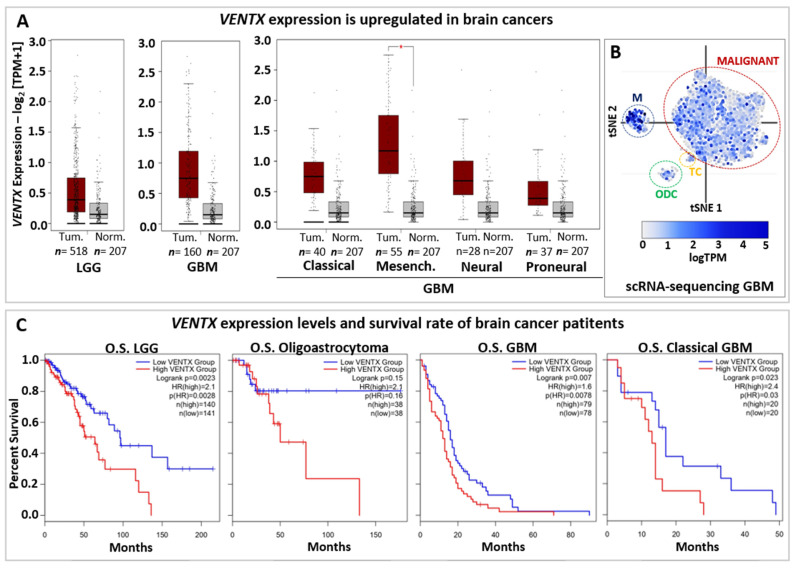
*VENTX* in brain cancers. (**A**) Expression of *VENTX* gene in human low-grade Glioma (LGG) and Glioblastoma (GBM). Primary tumors (Tum.; Red box) and normal samples (Norm.; grey box) from the TCGA and the GTEx datasets has been analyzed by using gene expression profiling interactive analysis (GEPIA2 database, http://gepia2.cancer-pku.cn, accessed on 1 July 2022) web resource. Note that *VENTX* expression is upregulated in both LGG (*n* = 518 samples) and GBM (*n* = 160 samples). Furthermore, high expression of *VENTX* is observed in GMB sub-types such as Classical, (*n* = 40 samples), Mesenchymal (Mesench.; *n* = 55 samples) and Neural (*n* = 28 samples). For statistical analysis, |Log_2_FC| Cutoff: 1 and *p*-value Cutoff: 0.01 have been used, * *p* < 0.01 (**B**) Single-cell RNA sequencing (scRNA-seq) expression distribution of *VENTX* in glioblastoma tumor mass from Broad Institute Single Cell Portal database (https://singlecell.broadinstitute.org/single_cell at 1 July 2022), GEO: GSE131928. *VENTX* is expressed in malignant tumor cells (MALIGNANT, red circle), Macrophages (M, blue circle), T-cells (TC, yellow circle) and oligodendrocytes (ODC, green circle). (**C**) Overall survival (OS) analysis on *VENTX* expression levels in human low-grade Glioma (LGG) and LGG subtype Oligoastrocytoma, Glioblastoma (GBM), and GBM subtype Classical GBM. Red line represents VENTX “high” expression, blue line represents VENTX “low” expression (Median Cutoff). Note that high *VENTX* expression significantly correlates (for LGG and GBM) with lower survival rate.

**Table 1 cells-11-02299-t001:** CRISPR/CAS9 modulation of VENTX in human from BioGRID Open Repository for CRISPR Screen (ORCS) database.

Cell Type	Cell Line	Phenotype	Author (Year)	PMID (NCBI)
**Glioma**	HS-683	cell proliferation	Meyers RM (2017)	29083409
**Glioblastoma**	G549NS (patient-derived)	cell proliferation	MacLeod G (2019)	30995489
**Neural Stem Cell**	HF7450 (primary-derived)	cell proliferation	MacLeod G (2019)	30995489
**Pancreatic Cancer**	PANC-1	response to chemicals	Ramaker RC (2021)	34049503
**Pancreatic Adenocarcinoma**	HPAF-2	cell proliferation	Steinhart Z (2017)	27869803
**Chronic Myeloid Leukemia**	K-562	cell proliferation	Liu J (2019)	31316073
**Renal Cell Carcinoma**	RCC 786-O	response to chemicals	Zou Y (2019)	30962421
**Non-Small Cell Lung Adenocarcinoma**	A549	response to chemicals	Gobbi G (2019)	31406246
**Non-Small Cell Lung Adenocarcinoma**	A549	cell proliferation	Gobbi G (2019)	31406246
**Ovarian Cancer**	TOV-21G	cell proliferation	Meyers RM (2017)	29083409
**Ovarian Cancer**	PEO1	cell proliferation	Wheeler LJ (2019)	31437751
**Urinary Bladder Cancer**	MGH-U4	response to chemicals	Goodspeed A (2019)	30414698

Data available on: https://orcs.thebiogrid.org/Gene/27287, accessed on 1 July 2022.

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
