# Peer review of "Vertebrate Cell Differentiation, Evolution, and Diseases: The Vertebrate-Specific Developmental Potential Guardians VENTX/NANOG and POU5/OCT4 Enter the Stage"

_cells, 2022, doi:10.3390/cells11152299_

Round 1

Reviewer 1 Report

In this review, Ducos et al discuss the diverse function and molecular mechanisms linked to pluripotency factors in early embryos and stem cells. they also discuss the implication of these factors in neural crest and NMPs populations of multipotent cells that emerge later on during development.  Finally, they discuss the implication of these genes in cancer. I believe this is an exciting review, which will be helpful in the field. In particular, the functions of pluripotency factors outside of stem cells is a puzzling observation that will probably lead to passionating research in the future.

I have a suggestion to try to improve the manuscript, and more importantly some comments about the NMPs section.

 - I think it would be great to have a more exhaustive list (table) of the cell types that re-express or maintain these factors, you mentioned hematopoiesis in some points, I also think about primordial germ cells, maybe other cell types? 

-    About NMPs I was confused about the fact that the authors focus first on Xenopus and zebrafish literature for describing DPG and NMPS shifting later  to the mouse model only to discuss Oct4 functions. I think a lot of discoveries/characterizations about NMPs have been made in mice and birds models and several of the seminal papers from the field have been missed. For instance, Cambray 2002 for characterization of NMPs at the tissue level,  Tzouanacou 2009 for the initial discovery of NMP cells, and Olivera and Martinez 2012 for coexpression of Sox2 and Bra. Even concerning the Zebrafish one critical paper Attardi et al 2018 is not cited and instead some single cells transcriptomic papers (not lineage) or reviews (which I think should be avoided if possible considering that the manuscript is already a review) are cited.  

Reviewer 2 Report

This is a very interesting review because it explains in details several complex developmental transitions that are difficult for non-embryologist to grasp.  Authors also propose several interesting hypotheses and several potential lines of research.

Unfortunately, the manuscript is not well written. There are many imprecisions, grammatical and English mistakes (plurals and tense accords, unnecessary or missing articles and words) that should be corrected by a native English speaker if possible.

In addition, there is quite a few conceptual issues in the manuscript that could be improved.

Figure 1b is unclear: what is controlling increased MEK1 activity during transition between refractory and competent states? Refractory and competent cells seem to receive the exact same signals ?

Graph should be made clearer.

Conceptually unclear why authors state that NEBs have a lower broader developmental potential than one of their derivatives the NCCs.  What is the definition of potency? DPGs have multiple roles: they are active in early development and are reactivated later in development, notably at the neural crest stage, but NCCs cannot have more potency than the cells they are coming from. This is a conceptually confusing. Rather, I would say that this example demonstrates that cells not expressing the DPGs can have a higher potency than their DPG-expressing derivatives.

In other words, placing the purple cells lower than the NCC in figure 2C is arbitrary. Defining developmental potency uniquely in reference to expression of DPGs is confusing because the terms has been used much more broadly for decades in the literature.

Maybe calling the VENTX/NANOG and POU5/OCT4 genes "Developmental Potency Guardian" is not such a good idea because of its broadness. These genes do control some form of developmental potency, but they are not the only genes that control "developmental potency" which is a very generic term that can be applied to any differentiation steps during development and many of these steps are not controlled by these genes.  Maybe a different, more precise acronym would be more pertinent and hep eliminate the confusion in the manuscript about potency.

Line 277: Why are NEB unipotent since they give rise to multiple lineages in vertebrates?. If I understand correctly, the authors mean to say that NEB are unipotent in invertebrates and multipotent in vertebrates because some ancestors of the vertebrates has acquired the ability to reactivate the DPG genes. Confusing to mix up evolutionary change and differentiation concept in the same sentence.  Sentence should be clarified

Line 330 and on:  A figure illustrating the location of the NMPs and other sites of expression of the DPG would be useful.

Lines 475 to 512: optogenetic approach is really interesting but description is too long and  to technical in the setting of a conclusion. Paragraphs should be considerably shortened.

TITLE : The Foundation of Vertebrate Cell Potency and Heterogeneity  in Development, Evolution and Diseases.

The title is too broad and does not reflect content of the review. This lack of specificity  might decrease the visibility of the review:

This review is focused on the role of DPG in differentiation, not on cell potency in general.

Title should reflect that

Round 2

Reviewer 1 Report

 I thank the authors for taking into account my concerns, the changes they made satisfied my concerns , and now I for me, this manuscript is suitable for publication.

I just want to point out two typos l 437 and 444: I guess the authors mean to say "amniotes versus non-amniotes"

Reviewer 2 Report

Very interesting review. Authors have addressed my previous concerns